# The Influence of Residual Sodium on the Catalytic Oxidation of Propane and Toluene over Co$_3$O$_4$ Catalysts

**Guangtao Chai [1,2], Weidong Zhang [1], Yanglong Guo [2,*], Jose Luis Valverde [3] and Anne Giroir-Fendler [1,*]**

[1] Université Claude Bernard Lyon 1, Université de Lyon, CNRS, IRCELYON, 2 Avenue Albert Einstein, F-69622 Villeurbanne, France; guangtao.chai@ircelyon.univ-lyon1.fr (G.C.); weidong.zhang@ircelyon.univ-lyon1.fr (W.Z.)

[2] Key Laboratory for Advanced Materials, Research Institute of Industrial Catalysis, School of Chemistry and Molecular Engineering, East China University of Science and Technology, Shanghai 200237, China

[3] Department of Chemical Engineering, University of Castilla La Mancha, Avda. Camilo José Cela 12, 13071 Ciudad Real, Spain; jlvalverde1964@gmail.com

\* Correspondence: ylguo@ecust.edu.cn (Y.G.); anne.giroir-fendler@ircelyon.univ-lyon1.fr (A.G.-F.); Tel.: +86-21-64-25-29-23 (Y.G.); +33-472-431-586 (A.G.-F.)

**Abstract:** A series of Co$_3$O$_4$ catalysts with different contents of residual sodium were prepared using a precipitation method with sodium carbonate as a precipitant and tested for the catalytic oxidation of 1000 ppm propane and toluene at a weight hourly space velocity of 40,000 mL g$^{-1}$ h$^{-1}$, respectively. Several techniques were used to characterize the physicochemical properties of the catalysts. Results showed that residual sodium could be partially inserted into the Co$_3$O$_4$ spinel lattice, inducing distortions and helping to increase the specific surface area of the Co$_3$O$_4$ catalysts. Meanwhile, it could negatively affect the reducibility and the oxygen mobility of the catalysts. Moreover, residual sodium had a significant influence on the catalytic activity of propane and toluene oxidation over the synthesized Co$_3$O$_4$ catalysts. The catalyst derived from the precursor washed three times presented the best activity for the catalytic oxidation of propane. The origin was traced to its better reducibility and higher oxygen mobility, which were responsible for the formation of active oxygen species. On the other hand, the catalyst obtained from the precursor washed two times exhibited better performance in toluene oxidation, benefitting from its more defective structure and larger specific surface area. Furthermore, the most active catalysts maintained constant performance in cycling and long-term stability tests of propane and toluene oxidation, being potentially applicable for practical applications.

**Keywords:** catalytic oxidation; propane; toluene; Co$_3$O$_4$; residual sodium

## 1. Introduction

Volatile organic compounds (VOCs), mainly released from transportation and industrial processes, cause serious environmental pollution, and have toxic effects on human health [1,2]. They contribute to the formation of photochemical smog and further promote the production of ozone. Moreover, the degradation of VOCs is usually difficult and easily produces secondary pollutants, resulting in long-lasting pollution and impacting the ecosystem. Thus, the removal of VOCs is of great importance to protect the environment [3–6]. Over the past decades, research has focused on developing different efficient techniques to degrade VOCs. One of the most promising and economical techniques to remove these pollutants is the catalytic oxidation method, which allows VOC removal from dilute streams at a

low temperature with high efficiency. With this method, VOCs can be completely converted into $CO_2$ and $H_2O$ [7–11].

Noble metal (Pt, Pb, Au, Ru, etc.) dispersed on supports with high specific surface areas have long been studied because of their remarkable performance in VOC oxidation [12–15]. However, low dosages of noble metals are required while maintaining the activity due to their high price and low availability. Moreover, it is important to prevent the agglomeration of metal particles in practical applications to reduce the tendency towards deactivation and prolong the lifetime. Consequently, the use of catalysts based on transition metal (Fe, Co, Ni, Mn, Cu, Ce, etc.) oxides has been explored as an alternative approach [16–20].

Spinel cobalt oxide, $Co_3O_4$, with $Co^{2+}$ surrounded by a tetrahedral $O^{2-}$ coordination sphere and $Co^{3+}$ surrounded by octahedral $O^{2-}$ environment, has been proven to be one of the most promising and efficient materials for several catalytic reactions due to the advantages of low-temperature reducibility, high bulk oxygen mobility, and facile formation of active surface oxygen species [21–24].

The synthesis process is of great importance to $Co_3O_4$ catalysts since it could determine the physicochemical properties, such as the crystallite size, morphology, and reducibility of $Co_3O_4$ catalysts, and then affect their catalytic activity. Precipitation is one of the most widely used methods to prepare $Co_3O_4$ and other metal oxide catalysts in heterogeneous catalysis, which requires a simple set-up, short time, and can be easily scaled for practical industrial applications [25–28]. However, it should be noted that the complete removal of precipitant is difficult before and even after the calcination of catalyst precursors. Jun and co-workers [29] found that the residual sodium existed in the form of $NaNO_3$ after calcination and it decreased the reducibility of CuO phase in the $Cu/ZnO/Al_2O_3$ catalyst. Yu and co-workers [30] revealed that the residual sodium in the catalyst precursor was unfavorable to the catalytic performance and the stability of $Cu-CeO_2-Al_2O_3$ catalyst. Choya and co-workers [31] synthesized two $Co_3O_4$ samples via precipitation with and without residual sodium and found that the surface sodium decreased the $Co^{3+}/Co^{2+}$ molar ratio and mobility of active lattice oxygen species, leading to an overall negative impact on the catalytic activity of methane combustion. To the best of our knowledge, the influence of residual precipitants on the catalytic activity of $Co_3O_4$ catalysts for the oxidation of propane and toluene has not been reported yet.

In this work, a series of $Co_3O_4$ catalysts were synthesized via a precipitation method using sodium carbonate as the precipitant and labeled as Co-CO₃-xT, which will be described in detail in the section on catalyst preparation. The catalysts were investigated in the total oxidation of propane and toluene, respectively. All the catalysts were characterized by Thermogravimetric and Differential Scanning Calorimeter (TG-DSC), Inductively Coupled Plasma Optical Emission Spectroscopy (ICP-OES), X-ray Diffraction (XRD), Raman, $N_2$-sorption, Fourier Transform Infrared (FT-IR), and Carbon Monoxide-Temperature Programmed Reduction (CO-TPR). Propane oxidation without oxygen was evaluated as well. Moreover, cycling and long-term stability tests were carried out on the most active catalysts in the end. The aim of this work was to investigate the influence of residual sodium on the activity of $Co_3O_4$ catalysts, as well as the structure-performance relationship of the catalysts.

## 2. Results and Discussion

### 2.1. Decomposition of the Catalyst Precursors

TG-DSC analyses were carried out to investigate the thermal decomposition of cobalt oxide precursors. Similar weight loss curves were observed for all the precursors, as shown in Figure 1a. There was a small weight loss of ca. 2 wt.% below 200 °C, which was caused by the removal of physically and chemically bound water. In the temperature range of 200 to 300 °C, the curves showed an intense loss of mass (ca. 23 wt.%) together with an endothermic peak at around 255 °C due to the thermal decomposition of cobalt hydroxide carbonate in the catalyst precursors [23,32]. The endothermic peak in the DSC curves, as shown in Figure 1b, shifted to higher temperatures when increasing the number

of washing steps. Moreover, no obvious weight loss occurred above 400 °C, suggesting that 500 °C was enough for the formation of thermally stable $Co_3O_4$.

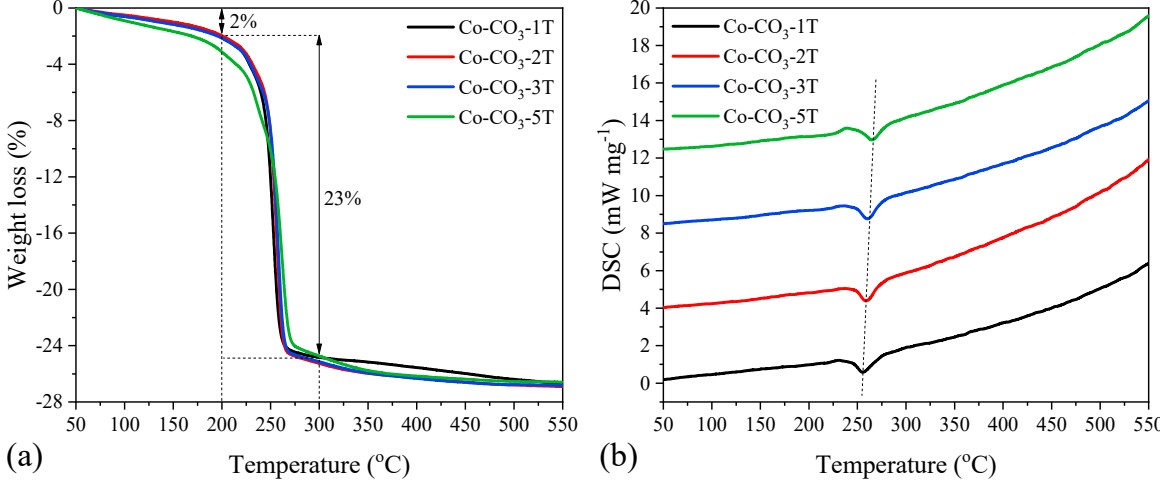

**Figure 1.** (**a**) TG and (**b**) DSC curves of the catalyst precursors.

### 2.2. Characterization of the Catalysts

The residual sodium content present in the catalysts was measured by ICP-OES. The results are shown in Table 1. This value was extremely high (2.4 wt.%) in sample Co-CO$_3$-1T, whereas it significantly decreased to 0.33 wt.% after washing the catalyst precursor twice, and further dropped to 0.14 wt.% with one more washing in the sample Co-CO$_3$-3T. In this sense, the sample Co-CO$_3$-5T, washed five times, was prepared in order to obtain a catalyst free of sodium in its structure.

**Table 1.** Results from physico-chemical characterizations.

| Catalyst | Na Content [a] (wt.%) | d [b] (nm) | a [b] (Å) | SSA [c] (m$^2$ g$^{-1}$) | Average Pore Size [c] (nm) | V$_{pore}$ [c] (cm$^3$ g$^{-1}$) |
|---|---|---|---|---|---|---|
| Co-CO$_3$-1T | 2.40 | 19.8 | 8.091 | 28 | 12.9 | 0.085 |
| Co-CO$_3$-2T | 0.33 | 30.8 | 8.088 | 27 | 11.8 | 0.081 |
| Co-CO$_3$-3T | 0.14 | 31.3 | 8.088 | 25 | 10.3 | 0.064 |
| Co-CO$_3$-5T | <0.1 | 33.0 | 8.088 | 22 | 8.8 | 0.049 |

[a] Determined by ICP-OES analysis. [b] Average crystallite size, lattice constant calculated from the XRD patterns.
[c] Specific surface area obtained using Brunauer-Emmet-Teller (BET) method, average pore size, and total pore volume obtained using Barrett-Joyner-Halenda (BJH) method.

Figure 2a displays the wide-angle XRD patterns of the catalysts with different residual sodium contents. Diffraction peaks at 2θ = 18.9, 31.2, 36.6, 44.9, 59.4, and 69.1° were observed and attributed to a cubic phase of spinel $Co_3O_4$ (PDF # 74-2120). These peaks could be ascribed to the (111), (220), (311), (400), (511) and (440) planes of $Co_3O_4$. Like other studies reported [31,33], no phases associated with sodium species were observed in any of the samples, possibly due to the good dispersion and low contents of sodium. However, the diffraction peaks became sharp and intense as the number of washing steps increased, suggesting the growth of crystallite size, which was confirmed by the results calculated from the most intense (311) diffraction peak given by the Scherrer equation (Table 1). Moreover, it could be seen that the (311) diffraction peak in the XRD pattern of sample Co-CO$_3$-1T appeared at a lower diffraction angle compared with that of other samples, as shown in Figure 2b, indicating that sodium may be introduced into the spinel lattice of $Co_3O_4$, inducing the structure distortion and inhibiting the crystallization of $Co_3O_4$ [31]. Taking as a reference the above-mentioned six diffraction peaks, the lattice constant of each sample was also estimated. This way, it was observed that the values of the lattice constant in Co-Co$_3$-2T, Co-Co$_3$-3T, and Co-Co$_3$-5T were identical, while that of Co-Co$_3$-1T was larger, which could be again associated with the presence of sodium into the $Co_3O_4$ lattice.

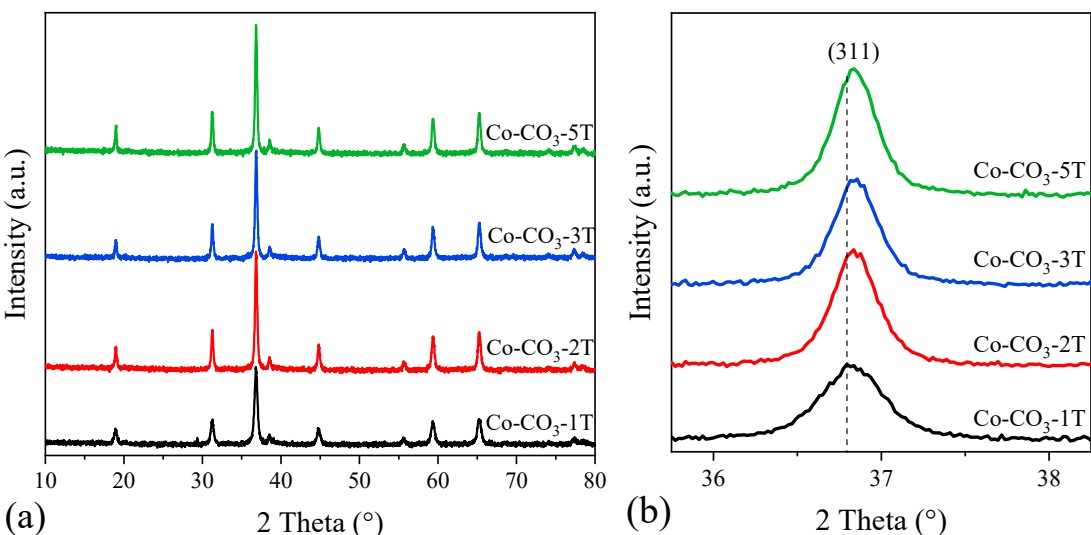

**Figure 2.** (**a**) XRD patterns of the catalysts; (**b**) enlargement of the most intense diffraction peak (311).

The Raman spectra allow us to get information about the lattice distortion of the catalysts (Figure 3). Five bands at 191, 474, 514, 609, and 679 $cm^{-1}$ were observed, corresponding to the stretching modes of the spinel $Co_3O_4$ [23]. The bands at 474 and 679 $cm^{-1}$ were attributed to the $E_g$ and $A_{1g}$ modes, respectively, whereas the bands at 191, 514, and 609 $cm^{-1}$ could be assigned to the $F_{2g}$ modes. $A_{1g}$ mode was associated with the octahedral sites, whereas the $F_{2g}$ and $E_g$ modes corresponded to the vibration of tetrahedral and octahedral sites [7,22]. It was found that the Raman bands of Co-CO$_3$-1T shifted to lower frequencies compared with those of other catalysts, which was due to the lattice expansion and lattice distortion caused by the introduction of sodium. This result agrees well with the XRD analysis. Moreover, the red shift was also observed in Co-CO$_3$-2T, which suggests the defective structure of the catalyst.

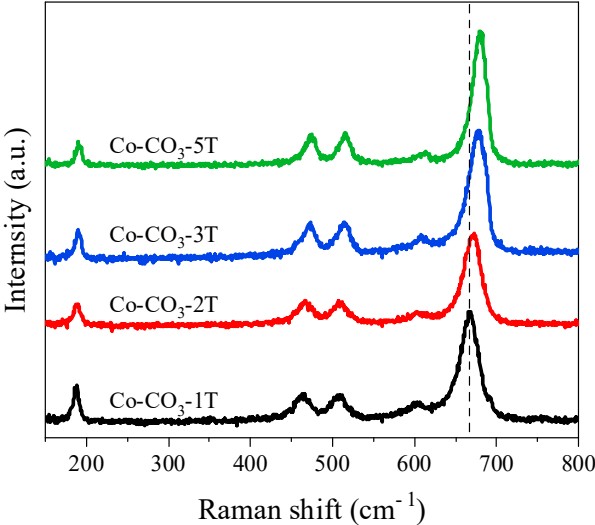

**Figure 3.** Raman spectra of the catalysts.

Adsorption bands at 3502, 3379, 1544, 1347, 1070, 969, 837, 779, 699, and 656 $cm^{-1}$ attributable to the stretching vibration modes of the cobalt hydroxide carbonate structure were observed in the FT-IR spectra of the catalyst precursors, as shown in Figure S1 [34,35]. The FT-IR adsorption spectra of the catalysts are shown in Figure 4a. Bands corresponding to the catalysts mainly occurred at 656 and 546 $cm^{-1}$, which was associated with the stretching vibration of Co–O in the $Co_3O_4$ spinel lattice. The former could be assigned to the stretching vibration mode of $Co^{2+}$–O from the tetrahedral

sites, whereas the latter was attributed to the stretching vibration of $Co^{3+}$–O in the octahedral coordination [32]. One can see in Figure 4b that the stretching vibration bands of octahedrally coordinated $Co^{3+}$ and tetrahedrally coordinated $Co^{2+}$ shifted to lower wavenumbers when the number of washing steps of the precursor was more than one, which would indicate the decrease in the strength of the Co–O bond [36]. In addition, some additional infrared adsorption bands at 836 and 1356 $cm^{-1}$ were clearly observed in the sample $Co-CO_3$-1T in Figure 4c, which could be assigned to the stretching vibration of the undecomposed nitrate groups [37].

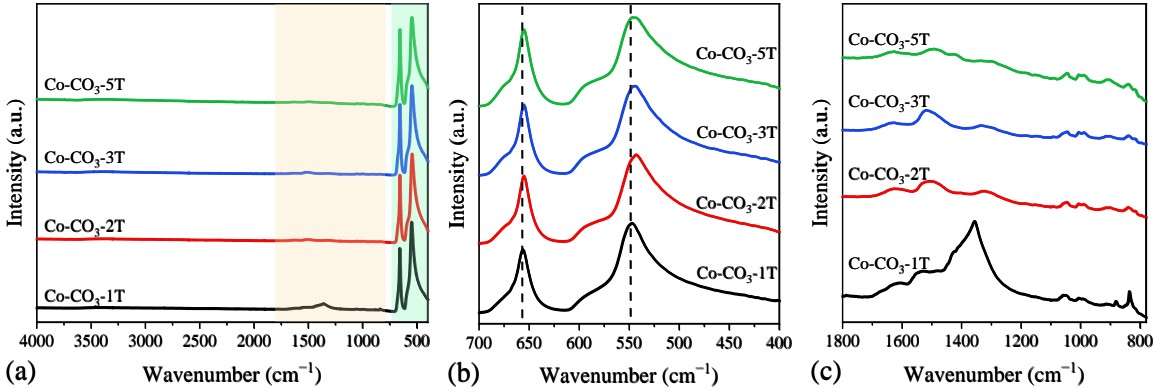

**Figure 4.** (**a**) Full and (**b**,**c**) partial FT-IR spectra of the catalysts.

Figure 5 shows the $N_2$-sorption isotherms and pore size distribution curves of the catalysts. All catalysts exhibited type IV isotherms with H3 hysteresis loops according to the International Union of Pure and Applied Chemistry (IUPAC) classification, being mesoporous in structure. The specific surface area (SSA), pore volume ($V_{pore}$), and average pore size of the catalysts are a function of (Table 1) the content of residual sodium. $Co-CO_3$-1T showed the largest SSA and $V_{pore}$, as well as the average pore size, which could be ascribed to the enhancement of textural properties caused by the distortion in the spinel lattice of $Co_3O_4$. The lower the residual sodium, the lower the values of SSA and $V_{pore}$ both were observed, suggesting that the $Co_3O_4$ particles could easily sinter during the thermal treatment with less residual sodium. Regarding the pore size distribution of the catalysts, it was found that all samples exhibited a similar pore size of ca. 2.5 nm. However, sample $Co-CO_3$-1T had a broader pore size distribution than other samples, which might be attributed to the presence of accumulated holes between particles by nano-$Co_3O_4$ crystallite bridges [36], which could, in turn, contribute to the increase of the pore volume.

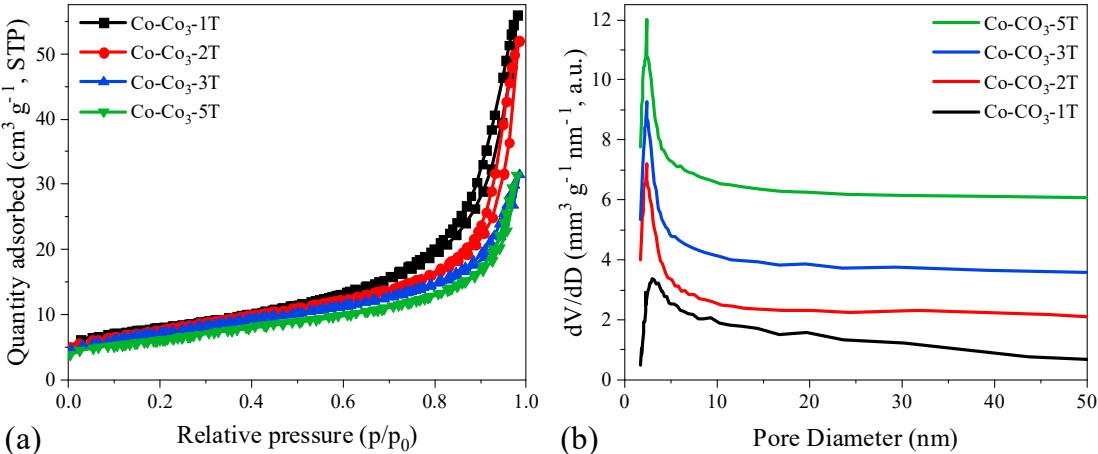

**Figure 5.** (**a**) $N_2$-sorption isotherms and (**b**) pore size distribution curves of the catalysts.

The CO-TPR experiment allows us to evaluate the reducibility of a catalyst. The evolutions of $CO_2$ concentration at the device outlet during the reduction tests are shown in Figure 6 and those of CO consumption are displayed in Figure S2. The consumption of CO matches perfectly with the formation of $CO_2$, indicating that all the consumed CO molecules were oxidized into $CO_2$ by depletion of the oxygen species in the catalysts. As reported in Reference [38], the reduction process started for all samples from around 200 °C and was completed below 600 °C, with two main reduction peaks observed. The former one in the range of 200 to 350 °C was associated with the reduction of $Co_3O_4$ to CoO; the latter one detected between 350 °C and 550 °C was assigned to the reduction of CoO to metallic Co. From the profiles of $CO_2$ concentration, one could see that the samples Co-CO$_3$-3T and Co-CO$_3$-5T were the most reducible ones with the lowest initial reduction temperature, followed by the sample Co-CO$_3$-2T. Catalyst Co-CO$_3$-1T showed the highest initial reduction temperature, which could probably be explained by the negative impact of residual sodium on the reducibility of $Co_3O_4$. It was reported that the reactivity of surface oxygen species and the mobility of lattice oxygen played a remarkable role in the oxidation performance of a catalyst [38,39], which would be reflected by the reducibility of the catalysts. Thus, the shift of the reduction peak to a lower temperature would indicate the presence of more active oxygen species on catalysts Co-CO$_3$-3T and Co-CO$_3$-5T. It should be pointed out that a small peak of $CO_2$ production appeared at 235 °C for sample Co-CO$_3$-1T, which could be attributed to the reduction of surface $NO_3^-$, as confirmed by the aforementioned FT-IR analysis.

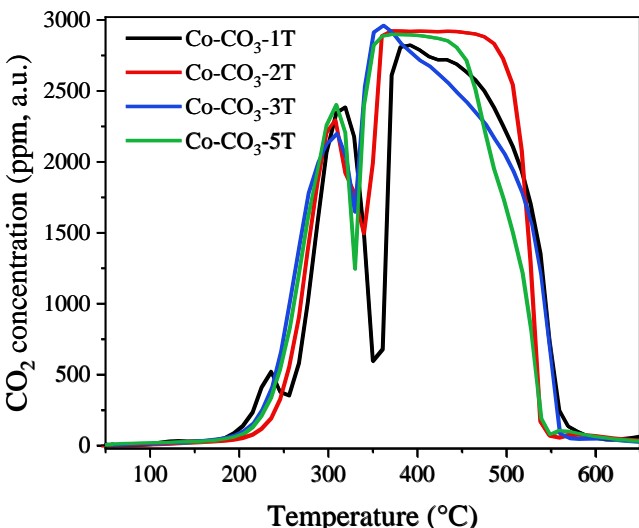

**Figure 6.** The evolution of $CO_2$ concentration during the CO-TPR tests of the catalysts.

### 2.3. Catalytic Activity

Propane oxidation was chosen as the model reaction for short-chain alkanes. The light-off curves of the third cooling run over the catalysts are plotted in Figure 7a. The $T_{50}$ and $T_{90}$ values (temperatures at conversions of 50% and 90%, respectively) in propane oxidation are summarized in Table 2. Catalyst Co-CO$_3$-3T exhibited the best performance for propane oxidation with $T_{50}$ and $T_{90}$ of 201 and 236 °C, respectively, which could be attributed to its highest reducibility. This catalyst was followed by catalyst Co-CO$_3$-5T, with similar activity in the high conversion range but a little worse performance when the reaction temperature was lower than 230 °C, probably due to the small SSA compared with Co-CO$_3$-3T. On the other hand, catalyst Co-CO$_3$-2T with lower reducibility presented lower catalytic performance (the $T_{90}$ value was 22 °C higher than that of catalyst Co-CO$_3$-3T). Finally, catalyst Co-CO$_3$-1T showed nearly no catalytic activity in the given temperature, which evidenced the strong inhibiting effect of excess residual sodium [31].

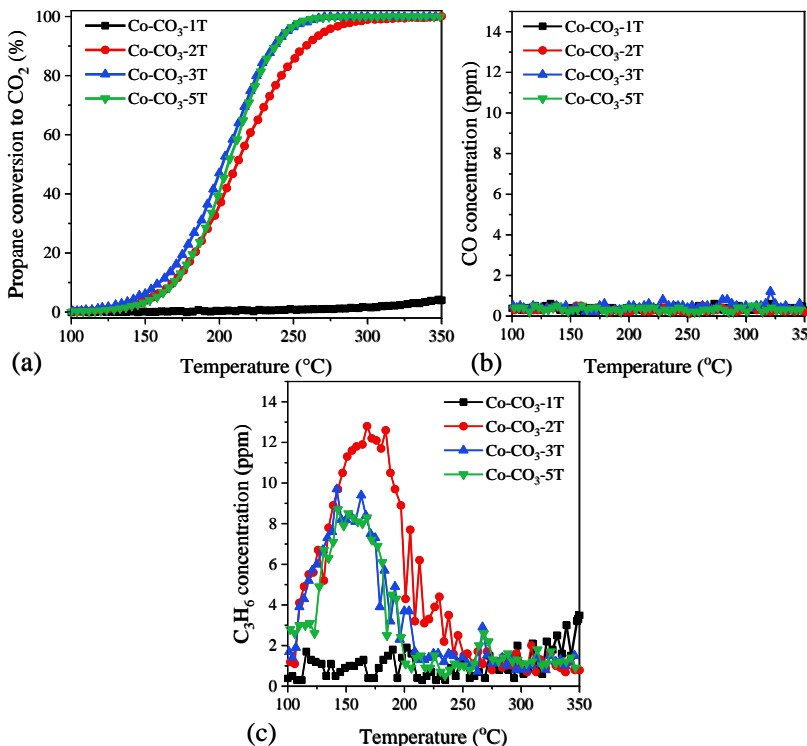

**Figure 7.** (**a**) Propane conversion to CO$_2$, (**b**) CO concentration, and (**c**) propene concentration as a function of the reaction temperature in the oxidation of propane over the catalysts.

**Table 2.** Catalytic performance and kinetic analysis of the catalysts for propane oxidation.

| Catalyst | T$_{50}$ (°C) | T$_{90}$ (°C) | r × 10$^8$ (mol g$^{-1}$ s$^{-1}$) | TOF × 10$^9$ (mol m$^{-2}$ s$^{-1}$) | E$_a$ (kJ mol$^{-1}$) | lnA |
|---|---|---|---|---|---|---|
| Co-CO$_3$-1T | / | / | 0.3 | 0.1 | / | / |
| Co-CO$_3$-2T | 211 | 258 | 6.6 | 2.4 | 80 | 4.8 |
| Co-CO$_3$-3T | 201 | 236 | 9.9 | 3.9 | 76 | 4.1 |
| Co-CO$_3$-5T | 206 | 238 | 6.9 | 3.1 | 92 | 8.2 |

The CO and propene concentrations at the reactor exit were in-situ recorded during the whole reaction process, as shown in Figure 7b,c to get a better understanding of the selectivity of the samples. It was found that CO was not produced with any of the catalysts tested, while traces of propene were detected in the temperature range of 100–200 °C for all catalysts except Co-CO$_3$-1T (inactive), suggesting propene was an intermediate product of propane oxidation over the Co$_3$O$_4$ catalysts. No obvious difference in propene production was observed, demonstrating the superior selectivity of Co$_3$O$_4$ catalysts for propane oxidation.

The propane oxidation reaction rates and TOF were calculated at 175 °C, and the results are listed in Table 2. Consequently, catalyst Co-CO$_3$-3T exhibited the largest reaction rate (9.9 × 10$^{-8}$ mol g$^{-1}$ s$^{-1}$) and the largest TOF value (3.9 × 10$^{-9}$ mol m$^{-2}$ s$^{-1}$). Apparent activation energies and apparent pre-exponential factors were obtained from the Arrhenius plots, as shown in Figure 8a, and the results are also displayed in Table 2. The pre-exponential factors followed the same trend as the E$_a$ did. Catalyst Co-CO$_3$-3T presented the lowest values of E$_a$ (76 kJ mol$^{-1}$), which was consistent with its best catalytic performance for propane oxidation.

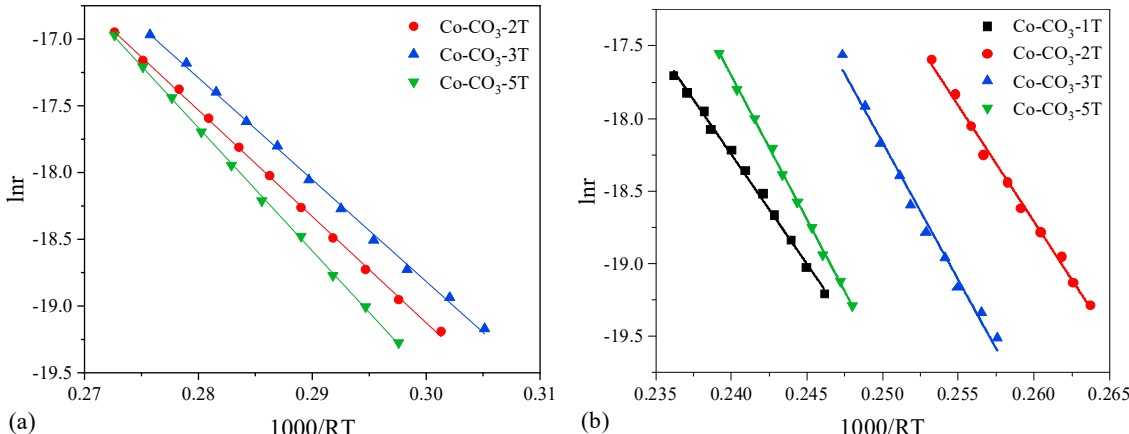

**Figure 8.** Arrhenius plots of lnr vs. 1000/RT for (**a**) propane oxidation and (**b**) toluene oxidation.

Propane oxidation without oxygen was carried out to examine the mobility of the oxygen species in the catalysts. Figure 9 shows the evolution of $CO_2$ concentration at the reactor exit during the tests. One could see that propane could be oxidized into $CO_2$ without the involvement of gaseous $O_2$ but with the help of oxygen species in the catalysts. The $CO_2$ productions over catalysts Co-CO$_3$-3T and Co-CO$_3$-5T were almost the same and were much higher than that over catalyst Co-CO$_3$-2T, whereas nearly no $CO_2$ was produced with catalyst Co-CO$_3$-1T, which agreed well with the aforementioned catalytic activity trend. Moreover, the surface lattice oxygen rather than the bulk lattice oxygen in the catalysts should be responsible for the $CO_2$ evolution considering the relatively low reaction temperature. The higher $CO_2$ production over catalysts Co-CO$_3$-3T and Co-CO$_3$-5T could be attributed to the better mobility of the surface lattice oxygen on them, which in turn contributed to their excellent performance in propane oxidation. Overall, the mobility of lattice oxygen and the reducibility of the samples played a remarkable role in the catalytic performance of the catalysts tested.

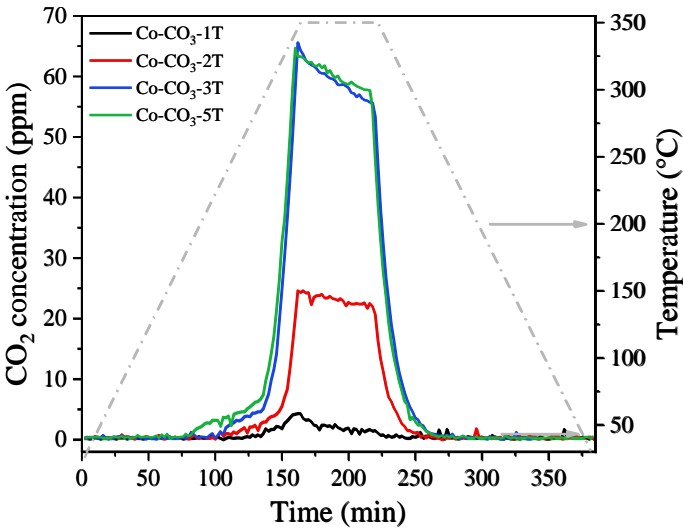

**Figure 9.** Evolution of $CO_2$ concentration in propane oxidation without oxygen over the catalysts.

Toluene was selected as the typical aromatic hydrocarbon for catalytic activity tests. Figure 10 shows the light-off curves of the catalysts in the third cooling run. The values of $T_{50}$ and $T_{90}$ are listed in Table 3. Different from that observed in the propane oxidation, the catalytic performance was ranked as follows: Co-CO$_3$-2T > Co-CO$_3$-3T > Co-CO$_3$-5T > Co-CO$_3$-1T. The most active catalyst, Co-CO$_3$-2T, exhibited excellent performance for toluene oxidation with $T_{50}$ and $T_{90}$ values of 232 and 248 °C, respectively, although it possessed worse reducibility than catalysts Co-CO$_3$-3T and Co-CO$_3$-5T.

This fact would indicate that the reducibility was not the key factor determining the catalytic activity of different catalysts in the toluene oxidation. Interestingly, catalyst Co-CO$_3$-1T, with nearly no activity in the given temperature range for propane oxidation, yielded a toluene conversion of 68% at 350 °C, suggesting that the inhibiting effect of residual sodium in Co$_3$O$_4$ was less severe on toluene oxidation than on propane oxidation. The largest SSA and highly defective structure of catalyst Co-CO$_3$-2T, which were related to more active sites and more surface adsorbed oxygen species [40], were supposed to be responsible for its high activity in toluene oxidation.

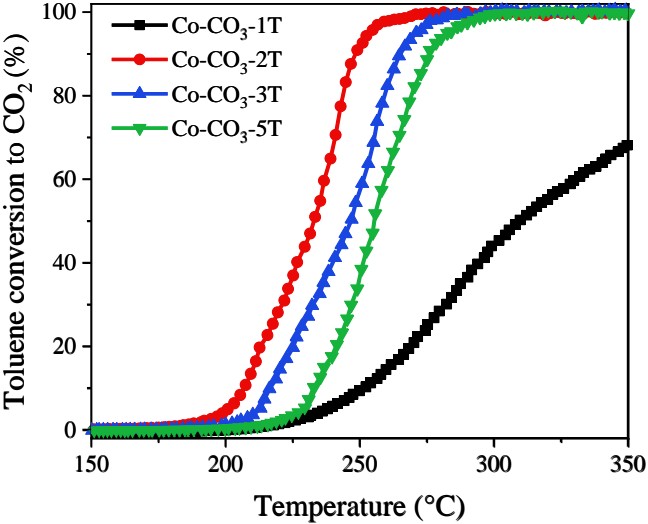

**Figure 10.** Light-off curves of the catalysts in the oxidation of toluene.

**Table 3.** Catalytic performance and kinetic analysis of the catalysts for toluene oxidation.

| Catalyst | $T_{50}$ (°C) | $T_{90}$ (°C) | $r \times 10^8$ (mol g$^{-1}$ s$^{-1}$) | $TOF \times 10^9$ (mol m$^{-2}$ s$^{-1}$) | $E_a$ (kJ mol$^{-1}$) | lnA |
|---|---|---|---|---|---|---|
| Co-CO$_3$-1T | 309 | / | 0.3 | 0.1 | 151 | 18.0 |
| Co-CO$_3$-2T | 232 | 248 | 10.0 | 3.7 | 160 | 22.8 |
| Co-CO$_3$-3T | 246 | 264 | 3.0 | 1.2 | 189 | 29.1 |
| Co-CO$_3$-5T | 255 | 276 | 0.5 | 0.3 | 198 | 29.9 |

The toluene oxidation reaction rates and TOF calculated at 213 °C for all catalysts are listed in Table 3. Catalyst Co-CO$_3$-2T with the best performance in toluene oxidation showed the largest value of reaction rate ($10.0 \times 10^{-8}$ mol g$^{-1}$ s$^{-1}$) and the largest TOF value ($3.7 \times 10^{-9}$ mol m$^{-2}$ s$^{-1}$). The Arrhenius plots were drawn for all catalysts, as shown in Figure 8b. The apparent activation energies and apparent pre-exponential factors obtained from the slopes and intercepts of the lines are also listed in Table 3. It could be observed that, apart from catalyst Co-CO$_3$-1T, catalyst Co-CO$_3$-2T presented the lowest values of $E_a$ (160 kJ mol$^{-1}$), agreeing well with its performance in toluene oxidation. Catalyst Co-CO$_3$-1T displayed the lowest $E_a$ but the excess residual sodium on the surface limited its activity towards toluene oxidation.

## 2.4. Stability Test

Three consecutive heating-cooling catalytic cycles were conducted to evaluate the cycling stability of the most active catalysts for propane and toluene oxidation. Figure S4 shows the three-cycle activity curves over the most active catalysts, namely Co-CO$_3$-3T for propane oxidation and Co-CO$_3$-2T for toluene oxidation. It can be seen that the three curves were almost the same for both propane and toluene oxidation, suggesting the good cycle-stability of these catalysts.

On the other hand, long-term durability tests over the most active catalysts for propane and toluene oxidation were also carried out (Figure 11). Regarding the propane oxidation, the test was performed as follows: the reactor was heated from room temperature to 350 °C and held at the latter

temperature for 1 h, then the temperature was decreased and controlled at 223 °C for 13.5 h and 193 °C for another 13.5 h. It was found that the conversion remained quite stable at both high and low conversion stages, demonstrating the good long-term stability of catalyst Co-CO$_3$-3T for propane oxidation. Regarding the toluene oxidation, the test was conducted by following the same procedure but with the temperature set at 245 °C for 24 h and 223 °C for 24 h, respectively. It was observed that the activity could be kept at the high conversion stage, whereas there was an obvious loss of activity at the low conversion stage. Considering the hysteresis phenomenon between the heating and cooling processes, where the hydrocarbon was completely converted into CO$_2$ in the heating run, as shown in Figure S3, the origin of the activity loss at the low conversion stage could be related to the formation of the carbonaceous species on the surface of the catalyst preventing the oxidation of toluene, which was also observed by other researchers [23,41].

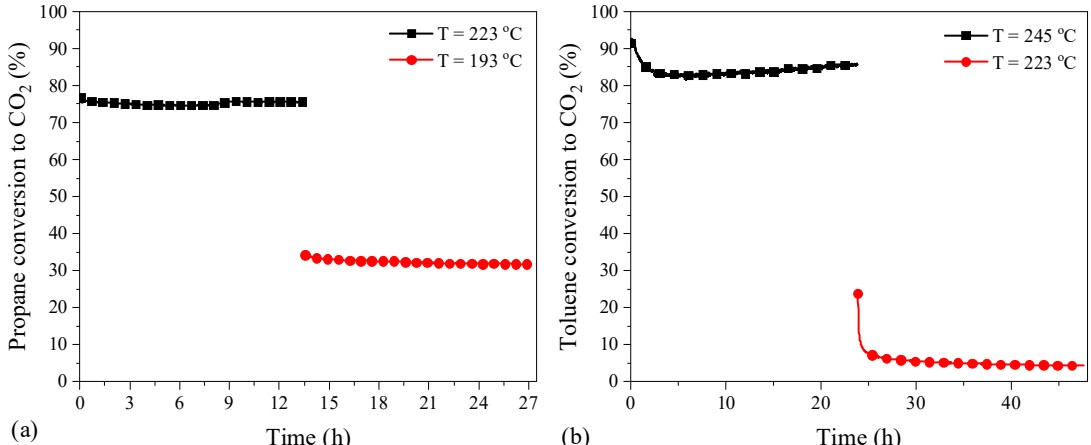

**Figure 11.** The influence of time on stream (**a**) propane and (**b**) toluene oxidation over the most active catalysts (cooling ramp).

*2.5. Discussion*

In this study, it was proved that both the physicochemical properties of the synthesized Co-CO$_3$-xT catalysts and their catalytic activities for propane and toluene oxidation were influenced by the residual sodium in the catalysts. The sodium could be inserted into the Co$_3$O$_4$ spinel lattice, inducing defective structures and leading to a smaller crystallite size and larger specific surface area of the Co$_3$O$_4$ catalysts. However, the residual sodium could meanwhile have a negative influence on the reducibility and oxygen mobility of the catalysts. As a result, the residual sodium significantly affected the catalytic performance of the Co$_3$O$_4$ catalysts.

For propane oxidation, catalyst Co-CO$_3$-3T presented the highest activity, with T$_{90}$ of 236 °C (weight hourly space velocity (WHSV) = 40,000 mL g$^{-1}$ h$^{-1}$). However, catalyst Co-CO$_3$-2T with a little more residual sodium performed much worse (T$_{90}$ = 258 °C), and catalyst Co-CO$_3$-1T with high sodium content displayed nearly no activity, which indicated that the residual sodium could be destructive to the catalytic activity of the Co$_3$O$_4$ catalysts for propane oxidation. Furthermore, the relatively worse activity of Co-CO$_3$-5T than that of Co-CO$_3$-3T suggested that a proper number of washing steps of the precipitate during the catalyst preparation was important to maintain good textural properties as well as considerable catalytic activity of the Co$_3$O$_4$ catalysts. The better activity of catalyst Co-CO$_3$-3T was determined by its better reducibility and higher oxygen mobility, which favored the formation of active oxygen species that play decisive a role in the catalytic reaction.

For toluene oxidation, the most active catalyst was Co-CO$_3$-2T (T$_{90}$ = 248 °C at a WHSV of 40,000 mL g$^{-1}$ h$^{-1}$), followed by catalyst Co-CO$_3$-3T (T$_{90}$ of 264 °C). Even catalyst Co-CO$_3$-1T, with significant residual sodium, could display certain activity to toluene oxidation. The above results showed that toluene oxidation was less sensitive to residual sodium poisoning than propane oxidation over the Co$_3$O$_4$ catalysts. However, excessive residual sodium still played a damaging role in the

catalytic oxidation of toluene over the $Co_3O_4$ catalysts. A small amount of residual sodium in the bulk of catalyst Co-CO$_3$-2T helped produce a defective structure and large surface area, which were related to more accessible active sites and more reactive surface adsorbed oxygen species that could facilitate the adsorption and activation of reactants, and thus accelerate the oxidation of toluene.

## 3. Experimental

### 3.1. Catalyst Preparation

A series of $Co_3O_4$ catalysts were prepared via precipitation using sodium carbonate as the precipitant. Typically, 20 mmol cobalt nitrate hexahydrate ($Co(NO_3)_2 \cdot 6H_2O$, Sigma-Aldrich, St. Louis, MO, USA) and 22 mmol sodium carbonate ($Na_2CO_3$, Sigma-Aldrich) were dissolved in 100 mL water, respectively. Then, the solution of precipitant was added into the solution of $Co(NO_3)_2 \cdot 6H_2O$ under continuous stirring and the mixed solution was kept stirring for 1 h at room temperature. After centrifugation, the precipitates were washed with deionized water 1, 2, 3, and 5 times (200 mL each time) in order to obtain samples with various contents of residual sodium. The pH of the liquor would be close to neutral after 3 times of washing. All the precipitates were dried in an oven at 80 °C overnight. After that, the dried powders were calcined under static air in a muffle furnace at 200 °C for 2 h, and then, 500 °C for 2 h, respectively. The obtained $Co_3O_4$ catalysts were denoted as Co-CO$_3$-xT, where x represents the numbers of washing.

### 3.2. Catalyst Characterization

TG-DSC analysis of the catalyst precursors (80 °C dried overnight) was carried out in flowing air (50 mL min$^{-1}$) at a heating rate of 10 °C min$^{-1}$ from 50 °C to 550 °C with a mass of 3–7 mg on a SETARAM Setsys Evolution 12 calorimeter (SETARAM, Caluire, France). An empty 70-mL aluminum pan was used as a reference.

ICP-OES was conducted on a HORIBA Jobin Yvon Activa instrument (HORIBA, Paris, France) to determine the catalyst composition. The catalysts were dissolved in a mixture solution of $H_2SO_4$ and $HNO_3$ and heated at 250 °C before the measurement.

XRD patterns of the catalysts were recorded using a Bruker D5005 diffractometer (Bruker, Karlsruhe, Germany) with a Cu Kα radiation (λ = 0.154184 nm) to characterize the structural properties of the catalysts. Samples were scanned in 2θ range from 10° to 80° with a step size of 0.02° and a scanning speed of 2 s per step. Phase identification was conducted by comparison with the Joint Committee on Powder Diffraction Standards (JCPDS) database cards. The crystallite size of $Co_3O_4$ was calculated using the Scherrer equation based on the six most intense peaks of (111), (220), (311), (400), (511), and (440).

Raman spectra were recorded using a HORIBA Jobin Yvon Raman instrument (HORIBA, Paris, France) with an excitation wavelength of 514 nm, 20 s exposure time, 5 accumulation, ×50 objective, and 1800 grating lines per mm. The catalysts were scanned from 150 cm$^{-1}$ to 800 cm$^{-1}$.

FT-IR tests were carried out on a PerkinElmer FT-IR C92712 spectrometer (PekinElmer, Norcross, GA, USA) to get information on the spinel structure of the catalysts. The spectrum was recorded in the range of 400–4000 cm$^{-1}$ with an instrument resolution of 1 cm$^{-1}$.

$N_2$-sorption testing of the catalysts was conducted at −196 °C on a Micromeritics TRISTAR II apparatus (Micromeritics, Norcross, GA, USA). Each catalyst was degassed at 300 °C for 3 h before the measurement. The specific surface area (SSA) of the catalysts was obtained according to the BET method. The total pore volume ($V_{pore}$) and the pore size distribution were calculated by the Barrett-Joyner-Halenda (BJH) method.

CO-TPR tests was conducted in a conventional flow system equipped with a thermal conductivity detector (SRA, Lyon, France). The catalysts were pre-treated at 300 °C for 1 h with a He flow. After cooling down to room temperature, a CO/He flow containing 3000 ppm CO at a flow rate of 100 mL min$^{-1}$ was introduced and the catalyst was heated from 50 °C to 700 °C with a heating rate of 5 °C min$^{-1}$.

### 3.3. Activity Test

Catalytic oxidation reactions were performed in a U-shaped quartz fixed-bed reactor (220 mm in length and 4 mm in internal diameter) at atmospheric pressure. Also, 150 mg of catalyst, mixed with an amount of silicon carbide (SiC) to get a fixed-bed height of 6 mm, were used for the catalytic reaction.

For propane oxidation, the feed gases containing 1000 ppm propane (0.1 vol.% propane + 21 vol.% $O_2$ + 79 vol.% He) were passed through the reactor with a flow rate of 100 mL min$^{-1}$, giving a weight hourly space velocity (WHSV) of 40,000 mL g$^{-1}$ h$^{-1}$. The reactor was firstly heated to 100 °C with a heating rate of 5 °C min$^{-1}$ and kept at this temperature for 0.5 h to get stabilization. Then, the temperature was increased to 350 °C in a step of 2 °C min$^{-1}$ and held for 1 h. After that, the temperature was decreased to 100 °C in the same step as that in the temperature-rise process. For each sample, three consecutive heating-cooling catalytic cycles were carried out. The reactants and products were analyzed online using a gas chromatograph (SRA, Lyon, France) equipped with two thermal conductivity detectors, one for the detection of $O_2$ and CO, and another for the detection of $CO_2$, $H_2O$, $C_3H_6$, and $C_3H_8$. The conversion of $C_3H_8$ was calculated using the following equation:

$$X_{C_3H_8}(\%) = \frac{[CO_2]_{out} - [CO_2]_{in}}{3[C_3H_8]} * 100 \tag{1}$$

where $[CO_2]_{out}$, $[CO_2]_{in}$ and $[C_3H_8]$ represent the outlet and inlet $CO_2$ concentrations, and the initial propane concentration, respectively.

For propane oxidation without oxygen, catalysts were pre-treated at 300 °C for 1 h with an He flow and then cooled down to room temperature in the same flow. The gas line was switched to a $C_3H_8$/He flow (400 mL min$^{-1}$) containing 1000 ppm $C_3H_8$. A heating-cooling run was performed in the temperature range from 30 °C to 350 °C with a ramp rate of 2 °C min$^{-1}$ and a 1 h plateau was set up at the highest temperature.

In toluene oxidation, the reactant gases containing 1000 ppm toluene, generated by passing an air flow through a pure toluene-containing U-shape tube which was placed in a 5 °C thermostatic bath, were fed into the reactor with a flow of 100 mL min$^{-1}$, corresponding to a WHSV of 40,000 mL g$^{-1}$ h$^{-1}$. The reactor was first heated to 150 °C with a heating rate of 5 °C min$^{-1}$ and kept for 0.5 h in order to stabilize the system. Then, the temperature was increased to 350 °C in a step of 2 °C min$^{-1}$ and held for 1 h. Next, the reactor was cooled down to 150 °C with the same step. For each sample, three consecutive heating-cooling catalytic cycles were also conducted. CO and $CO_2$ concentrations in the products were in-situ recorded by a Rosemount X-STREAM Gas Infrared Analyzer (Emerson Electric, Ferguson, MO, USA). The conversion of toluene was calculated as follows:

$$X_{C_7H_8}(\%) = \frac{[CO_2]_{out} - [CO_2]_{in}}{7[C_7H_8]} * 100 \tag{2}$$

where $[C_7H_8]$ represents the initial toluene concentration.

### 3.4. Kinetic Measurement

The kinetic data were measured in a fixed-bed reactor at atmospheric pressure. The linear velocity of the gas in the reactor and the particle size of the catalysts were appropriate to exclude the influence of diffusion. The reaction rate, r (mol g$^{-1}$ s$^{-1}$), was calculated using the following equation [42]:

$$r = X * V/M \tag{3}$$

where X and V are the conversion and the flow rate (mol s$^{-1}$) of propane or toluene, respectively, M is the weight of the catalyst (g).

The turnover frequency, TOF (mol m$^{-2}$ s$^{-1}$), was calculated as follows [42]:

$$TOF = X * V/S \tag{4}$$

where S is the surface area (m$^2$) of the catalyst.

The apparent activation energy, E$_a$, was obtained based on the Arrhenius formula.

$$\ln r = \frac{-E_a}{RT} + \ln A \tag{5}$$

where R is the ideal gas constant (8.314 J mol$^{-1}$ K$^{-1}$), T is the reaction temperature (K), and A is the apparent pre-exponential factor. E$_a$ can be inferred from the slope of the linear plot of ln r versus 1/RT. The conversion of propane or toluene to CO$_2$ considered in this analysis was always lower than 10%.

## 4. Conclusions

A series of Co$_3$O$_4$ catalysts with different contents of residual sodium was prepared using a precipitation method using sodium carbonate as the precipitant and then evaluated for the total oxidation of propane and toluene, respectively. The residual sodium influenced both the physicochemical properties and the catalytic performance of the Co$_3$O$_4$ catalysts. Residual sodium could be partially inserted into the spinel lattice, inducing distortions and resulting in smaller crystallite size and larger specific surface area of the Co$_3$O$_4$ catalysts. Furthermore, the reducibility of the catalysts was inhibited by the presence of residual sodium. For propane oxidation, trace residual sodium could poison the Co$_3$O$_4$ catalysts. The best performance was achieved on Co-CO$_3$-3T, which was traced to the better reducibility and higher oxygen mobility of Co-CO$_3$-3T. In terms of toluene oxidation over the Co$_3$O$_4$ catalysts, the negative effect of residual sodium was lower than that in propane oxidation. Co-CO$_3$-2T was the most active catalyst. The defective structure and larger specific surface area, as well as more surface adsorbed oxygen species on the catalyst, were responsible for the excellent performance. Long-term stability tests showed that the activity of catalyst Co-CO$_3$-3T was stable for propane oxidation, while catalyst Co-CO$_3$-2T could keep its activity at a high conversion level in toluene oxidation.

**Supplementary Materials:** The following are available online at http://www.mdpi.com/2073-4344/10/8/867/s1, Figure S1: FT-IR spectra of the catalyst precursors, Figure S2: Evolution of CO concentration during the CO-TPR test of the catalysts, Figure S3: Toluene conversion to CO$_2$ as a function of temperature over catalyst Co-CO$_3$-2T in a heating-cooling run, Figure S4: Three-cycle activity curves over (a) Co-CO$_3$-3T for propane oxidation and (b) Co-CO$_3$-2T for toluene oxidation.

**Author Contributions:** G.C. and W.Z. designed the experiments; G.C. performed the experiments; G.C. and W.Z. analyzed the data; G.C. prepared the manuscript and J.L.V., A.G.-F. and Y.G. revised and corrected the manuscript, supervised the Ph.D. students, and managed the research programs. All authors have read and agreed to the published version of the manuscript.

**Funding:** This work was financially supported by the National Key Research and Development Program of China (2016YFC0204300), the National Natural Science Foundation of China (21577035), the commission of Science and Technology of Shanghai Municipality (15DZ1205305) and Auvergne Rhone Alpes Region (project PAI 2019 LS 203067).

**Acknowledgments:** This work was financially supported by the University Claude Bernard Lyon 1, the CNRS, and the Auvergne Rhone Alpes Region (project PAI 2019 LS 203067). The China Scholarship Council are also acknowledged for the grants of G.C. and W.Z.

**Conflicts of Interest:** The authors declare no conflict of interest.

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
