# Peer review of "The Influence of Residual Sodium on the Catalytic Oxidation of Propane and Toluene over Co3O4 Catalysts"

_catalysts, doi:10.3390/catal10080867_

Round 1
Reviewer 1 Report
In this manuscript, the authors investigated the effect of Na in Co3O4 prepared by coprecipitation of Co nitrates with an aqueous solution of Na2CO3 on its properties and activity in catalytic oxidation of propane and toluene. The authors employed a series of characterization techniques, including ICP-OES, TG-DSC, XRD, Raman, N2-sorption, FT-IR, and CO-TPR to study the properties of the catalysts. Based on their results, the author claimed that residual sodium could inhibit the crystallization of Co3O4 and help to increase the specific surface area of the Co3O4 catalysts. It could also negatively affect the reducibility and the oxygen mobility of the catalysts. The catalyst with low concentration of Na (Co-CO3-3T) showed the highest activity in oxidation of propane, while Co-CO3-2T showed the highest activity in toluene oxidation. Concentration of Na in Co3O4 affected reducibility, oxygen mobility and number of defects in Co3O4. The results herein presented are of interest for the readers of Catalysts, which deserves publication.
Acceptance is recommended after addressing the following minor points:
1. At the beginning (Abstract, the last paragraph of the Introduction) it is necessary to insert definition of the catalysts studied in the manuscript. The labeling of the catalysts is described in the Chapter 3.1 “Catalyst preparation”, but it is too late.
2. Conditions of precipitation and washing should be described more precisely.
3. The changes in the extent of Co3O4 crystallization do not have to be caused by presence of Na only. Washing itself can also contribute to gradual crystallization of solid phase.
4. Line 140 and others: The use of “washing times of the precursors” is not accurate. I recommend “number of washing steps”.
5. Fig. 6: CO2 concentration meets the value of 3000 ppm. Is it possible, when initial concentration of CO is 1000 ppm?
6. In Fig. 9, the arrow indicating temperatures is missing.
7. I recommend to place Fig. 11 a) and b) into Supplementary.
Reviewer 2 Report
The paper presents the impact of residued Na in Co3O4 catalysts for propane and toluene oxidation. Each characterization is well written and depply discussed, and thus the paper is appropriate for the publication, after the following minor point is revised.
Abstract: The use of ”Co-CO3-xT” should be avoided, because the readers cannnot understand the meaning of the catalyst specisication. Abstract should be clearly described the scientific significance and should be completely understandable without checking the body text.
